# Eating Habits Related to Pregnancy, Body Perception, Attractiveness and Self-Confidence

**DOI:** 10.3390/healthcare12191932

**Published:** 2024-09-26

**Authors:** Wioleta Faruga-Lewicka, Wiktoria Staśkiewicz-Bartecka, Patrycja Janiszewska, Martina Grot, Marek Kardas

**Affiliations:** 1Department of Food Technology and Quality Evaluation, Department of Dietetics, Faculty of Health Sciences in Bytom, Medical University of Silesia in Katowice, 40-055 Katowice, Poland; wstaskiewicz@sum.edu.pl (W.S.-B.); mkardas@sum.edu.pl (M.K.); 2Department of Reproductive Health and Sexology, Department of Women’s Health, Faculty of Health Sciences in Katowice, Medical University of Silesia in Katowice, 40-055 Katowice, Poland; d201133@365.sum.edu.pl; 3Department of Human Nutrition, Department of Dietetics, Faculty of Health Sciences in Bytom, Medical University of Silesia in Katowice, 40-055 Katowice, Poland; martinagrot971@gmail.com

**Keywords:** prenatal diet, pregnancy nutrition, woman, psychological well-being in pregnancy, body image

## Abstract

Background: Pregnancy is a period of many changes in a woman’s life, including those related to eating habits, taking care of health and fitness, as well as esthetic and psychological considerations. Methods: The study was conducted between January 2021 and December 2022. A questionnaire was used to conduct the study, the questions of which concerned eating habits, as well as perceptions of one’s body and changes in appearance during pregnancy. Respondents filled out the questionnaire twice. The first survey was in the first two weeks after the doctor’s confirmation of pregnancy and after the end of pregnancy, up to a maximum of 2 months after delivery. The purpose of the study was to evaluate changes in women’s eating habits during pregnancy compared to before pregnancy to analyze the impact of pregnancy on women’s body perception and attractiveness and self-confidence; furthermore, the study aimed to analyze the body mass index (BMI) of women before pregnancy and to compare weight gain during pregnancy depending on the initial BMI, taking into account different age groups. The values of measurable parameters were presented using the arithmetic mean, median, dominant and standard deviation. Non-measurable parameters were presented using percentages. The Chi2 test of homogeneity was used to examine differences between groups. Results: During pregnancy, respondents mostly ate five meals a day, increased the amount of vegetables and fruits in their diets, and ate their meals without rushing (defined as slowly, calmly and without measuring the time left until the end of the meal break). Respondents reduced their coffee intake during pregnancy, while they increased their water intake to about 2 L a day. In the survey, about 2% of respondents followed a weight-loss diet during pregnancy. Respondents who were more likely to take body circumference measurements equally often monitored their daily energy intake. Conclusion: Women changed their eating habits during pregnancy to healthier ones compared to before pregnancy. The women’s eating habits and well-being were also not affected by their age. Also, there was no significant relationship between eating habits and well-being in pregnant women, regardless of their age. The study may assist medical staff in promoting healthier eating habits and tailoring psychological support, which is crucial for the well-being of pregnant women.

## 1. Introduction

Pregnancy is a period of many changes in a woman’s life, including those related to eating habits, taking care of health and fitness, as well as esthetic and psychological considerations [1].

Eating habits in women during pregnancy have important implications for fetal development, maternal functioning and recovery after pregnancy, as well as for the child’s risk of contracting diet-related diseases in adulthood. Eating habits are defined as characteristic and repeated behaviors carried out under the influence of the need for nutrients and the fulfillment of social and emotional goals. The balanced diet of a woman in the period before and during pregnancy is particularly important for the optimal development of her child and the course of pregnancy, as well as for the preservation of the mother’s health. The full coverage of energy and nutrient requirements is one of the important factors in avoiding intrauterine underdevelopment of the fetus [2,3,4,5,6].

Proper nutrition during pregnancy should therefore be characterized by the full coverage of energy and all essential nutrients while avoiding excessive intake. The best measure of proper energy intake is the maintenance of normal body weight before pregnancy and adequate weight gain during pregnancy [6,7].

Pregnant women’s eating habits can also be influenced by other factors such as doctor’s or midwife’s recommendations, morning nausea and general well-being [2,3,4,5,6,7].

Proper eating habits in pregnant women, according to the recommendations of Prof. Dorota Szostak-Węgierek [6], primarily include the consumption of three main meals and two snacks per day. They should cover the body’s full energy requirements, and contain essential nutrients important for fetal development, without excessive caloric intake [6,7]. It is estimated that in the first trimester of pregnancy, energy requirements are higher by about 150 kcal per day compared to the pre-pregnancy state, in the second trimester by about 360 kcal per day, and in the third trimester by 475 kcal per day. The final value depends on several factors such as the mother’s age, lifestyle, physical activity, type of work and baseline body mass index (BMI) [8,9].

During pregnancy, both underweight and overweight are risk factors for the development of the baby and the functioning of the mother’s body. An overweight or obese pregnant woman can lead to fetal macrosomia and a higher risk of metabolic syndrome in adulthood. For pregnant women, it is associated with the risk of gestational diabetes and complications during pregnancy and childbirth [10]. For this reason, it is necessary to reduce the intake of sweets and limit the amount of animal fats and cholesterol-containing products in the diet [11].

An important element for women planning pregnancy and those who are pregnant is normal weight gain.

If a woman was underweight before pregnancy (BMI < 18.5), the recommended weight gain is 12.5–18 kg. For normal weight (BMI 18.5–24.9), the recommended gain is 11.5–16 kg, for overweight (BMI 25–29.9) 7–11.5 kg and for obese (BMI ≥ 30) 5–9 kg [7,11].

In the human subconscious, a woman’s attractiveness is influenced by her weight, size and body shape. In addition, in women, waist-to-hip ratio and weight predict not only attractiveness, but also health [12]. Pregnancy is a period of many changes in a woman’s body, which can often be unacceptable to herself [13]. In the literature, this period in a woman’s life is often referred to as a “psychological burden”, especially the first pregnancy [14]. It involves cognitive and emotional changes, and can be a stressor of great intensity [13].

In addition, social pressure imposed by social media depicting pregnant celebrities with ideal shapes makes one more susceptible to negative perceptions of one’s body [15]. The development of the media and the availability of information has made preoccupation with one’s own appearance increasingly important also during pregnancy and the postpartum period [16,17,18]. Adequate education for women about body image during and after pregnancy and support throughout this period can be a key element in strengthening pregnant women’s mental health [19,20].

The purpose of the study is to evaluate changes in women’s eating habits during pregnancy compared to before pregnancy in order to analyze the impact of pregnancy on women’s body perception and attractiveness and self-confidence. Another aim is to identify whether women’s age is a determinant of the above variables and to analyze the body mass index (BMI) of women before pregnancy and to compare weight gain during pregnancy depending on the initial BMI, taking into account different age groups.

## 2. Material and Methods

### 2.1. Study Design

#### 2.1.1. Subjects

The study was conducted between January 2021 and December 2022. A CAWI (Computer-Assisted Web Interview) method was used to obtain the results, a method in which the form was made available electronically [21]. A questionnaire was used to conduct the study, the questions of which concerned eating habits, as well as perceptions of one’s body and changes in appearance during pregnancy. Respondents filled out the questionnaire twice according to the following criteria to ensure reliable answers regarding the aspects studied:(1)The first survey in the first two weeks after the doctor’s confirmation of pregnancy, with all patients completing the questionnaire by the 12th week of pregnancy,(2)The second survey after the end of pregnancy, up to a maximum of 2 months after delivery.

#### 2.1.2. Data Collection

The Google Forms platform was used for data collection. To avoid the potential risk of questionnaires being filled out twice by the same person, the uniqueness of respondents was monitored by checking email addresses. To avoid the phenomenon of respondent bots, the necessity to answer all questions was applied.

The first questionnaire was made available to participants by posting a link to the survey in groups on the social networking platform Facebook. The groups were carefully selected to obtain a representative survey sample. Women who completed the first questionnaire correctly and met the inclusion and exclusion criteria were informed via email address about the second stage of the study. They were individually sent a link to the second questionnaire with a request to complete it. A diagram of the data collection process is shown in Figure 1.

#### 2.1.3. Ethical Consideration

The study used dedicated sampling. With this method, the sample was selected to represent characteristics, specific experiences and features related to the topic of the study. Defining precise selection criteria allowed the study to achieve its objectives.

Participants were informed about the purpose of the study, voluntary participation, anonymity and how personal data would be collected and processed. In addition, participants were asked to agree to participate in the study and to accept the rules for data sharing. Information about voluntary and informed participation in the study was included at the beginning of the questionnaire to ensure transparency and fairness in the research process. The Declaration of Helsinki of the World Medical Association guided the conduct of this study. The study was approved by the Bioethics Committee of the Silesian Medical University in Katowice (PCN/0022/KB/299/19/20, date of approval: 29 January 2020) in light of the Law of 5 December 1996 on the Profession of Physician and Dentist, which includes a definition of medical experimentation.

### 2.2. Research Tools

A survey questionnaire was used to conduct the study, which consisted of a metric (data of the subject: age, education and anthropometric data—height and weight), a section that was created based on the WRB-Q (Weight Related Behaviours Questionnaire), developed by Kendall A, Olson C.M and Frongillo E.A [22]. Cronbach’s alpha of the scales ranged from 0.73 to 0.89.

### 2.3. Measurements

#### 2.3.1. WRB-Q Questionnaire

For the WRB-Q questionnaire, questions on the need to control one’s weight during pregnancy (question #1–4) and well-being, and perception of one’s own body (question #13–25) were used. The answers to the questions used a 5-point Likert scale from “strongly agree” to “strongly disagree”, where the third option corresponded to “neither agree nor disagree”.

#### 2.3.2. Eating Habits Questionnaire

The eating habits section of the questionnaire for women before and during pregnancy was developed based on the recommendations of Prof. Dorota Szostak-Węgierek, a recognized expert in the field, and WHO recommendations on antenatal care for a positive pregnancy experience [6,7]. The questionnaire aims to assess various aspects of dietary habits that may influence maternal health and pregnancy outcomes. It includes questions regarding meal frequency, timing of meals, the consumption of essential food groups like vegetables and fruits, hydration status, the use of dietary supplements and the intake of potentially unhealthy foods and beverages. Additionally, it addresses the practice of eating meals in a rushed manner or on the go, as well as the frequency of consuming sweets. This tool provides valuable insights into participants’ eating behaviors and helps researchers understand how these behaviors may impact maternal and fetal health during pregnancy.

#### 2.3.3. BMI Calculation

The subjects’ nutritional status was assessed using a body mass index calculated using the following formula: BMI (kg/m^2^) = body weight (kg)/height (m)^2^. Subsequently, the results were interpreted according to the World Health Organization (WHO) guidelines, where a BMI value ≥ 30.00 kg/m^2^ indicates obesity, values of 25.00–29.99 kg/m^2^ indicate overweight, 18.50–24.99 kg/m^2^ indicates normal weight and values of 17.00–18.49 kg/m^2^ indicate underweight [23].

### 2.4. Study Inclusion and Exclusion Criterion

The inclusion criteria for the study were as follows: informed and voluntary consent to participate in the study, 18 years of age, completion of the study questionnaire within 2 weeks of physician confirmation of pregnancy, pregnancy up to 12 weeks, normal course of pregnancy, no chronic diseases in the subject, properly completed study questionnaires.

The exclusion criteria for the study were as follows: failure to refill the questionnaire, refilling the questionnaire more than 8 weeks after the end of pregnancy, and finding malformations and diseases in children born up to 8 weeks after birth.

### 2.5. Statistical Analysis

Statistical analysis was performed using Statistica 13.3 software (TIBCO Software Inc., Palo Alto, CA, USA). The values of measurable parameters (e.g., measurement results) were presented using the arithmetic mean (average), median, dominant and standard deviation. Non-measurable parameters (e.g., qualitative scale scores) were presented using percentages. The Chi2 test of homogeneity was used to examine differences between groups.

A value of *p* < 0.05 was taken as the level of statistical significance.

## 3. Results

### 3.1. Sample Characteristics

The study group consisted of 113 women, but taking into account the inclusion and exclusion criteria, the target group included 101 women. Of the respondents, 79.2% (n = 80) of the women had a university education, and 20.80% (n = 21) had a high school education. Most respondents resided in urban areas (n = 71, 70.29%), and their material conditions were very good (30.69% of respondents), good (59.4%), or average (9.9%). The age of respondents was divided into two age categories: aged 18–30—55.45% of the respondents (n = 56), and aged 31–49—44.55% of the respondents (n = 45). The average age of those qualified for the study was 30.01 ± 4.75, and the height was 165.52 ± 5.29 cm. The mean weight of the women before pregnancy was 63.4 ± 13.23, while the mean weight in the third trimester of pregnancy was 77.4 ± 13.93. The characteristics of the study group are shown in Table 1.

### 3.2. Eating Habits

To evaluate the change in eating habits between the periods before and during pregnancy, the study utilized statistical comparisons of various dietary behaviors. The results indicate that there were significant changes in the eating habits of the respondents over the two time points. Specifically, there was a significant increase in the number of women eating five meals a day during pregnancy (73.27%) compared to before pregnancy (36.63%) (*p* < 0.01). Similarly, the regular consumption of breakfast increased from 52.47% before pregnancy to 88.12% during pregnancy (*p* < 0.00). The daily consumption of vegetables also showed a significant increase from 58.42% to 80.20% (*p* < 0.00), and the consumption of fruits rose from 58.42% to 84.16% (*p* < 0.00). Additionally, the number of women drinking at least 2 L of water daily significantly increased from 35.64% before pregnancy to 68.32% during pregnancy (*p* < 0.00). There was also a significant reduction in the consumption of fast food and processed foods, with 64.36% of women limiting these foods during pregnancy compared to 48.51% before pregnancy (*p* < 0.00). Finally, there was a marked improvement in food hygiene practices, with more women eating their meals in a calm and unhurried manner during pregnancy (87.13%) compared to before pregnancy (41.58%) (*p* < 0.00). A comparison of the eating habits of women before pregnancy and during pregnancy is shown in Table 2.

### 3.3. BMI before Pregnancy

The study analyzed women’s pre-pregnancy BMI and determined normal weight gain. Underweight women accounted for 12.9% (n = 13), normal BMI 55.45% (n = 56), overweight 22.77% (n = 23) and obese 8.91% (n = 9). The results are shown in Table 3.

### 3.4. Self-Assessment of Psychological Aspects of the Pregnancy Period among Women

Women in the preconception period at the highest level did not take frequent body measurements and did not calculate the energy requirements of their diet (n = 53, 52.47%). The comparison of the level of frequency of measurements taken with the monitoring of the energy requirements of the diet did not show a statistical significance (*p* = 0.14).

Respondents were asked about their well-being, attractiveness and self-confidence during pregnancy. Self-esteem was very bad and bad for 30.69% (n = 31) of the respondents, and good and very good for 45.54% (n = 46) of the women. Their attractiveness was described as bad and very bad by 33.66% (n = 34) of female respondents, and very good and good by 41.58% (n = 42). Low self-confidence (very bad and bad variants) was indicated by 28.71% of respondents (n = 29), and high confidence (good and very good variants) by 50.49% (n = 51) of women.

Women were also asked about their perception of the pregnancy period. A total of 72.22% (n = 78) of respondents specified that pregnancy was a positive period in their lives.

The highest percentage were women with their first pregnancy (n = 63; 62.38%), who rated their level of self-confidence as “well” to the greatest extent (n = 18; 28.57%). The number of past pregnancies compared with the sense of confidence scale (1–5) showed no statistical relationship (*p* = 0.98). Detailed data on psychological aspects during pregnancy are shown in Table 4.

## 4. Discussion

In the present study, we analyzed changes in women’s eating habits during pregnancy compared to before pregnancy. Also, we analyzed the impact of pregnancy on women’s body perception and attractiveness, and self-confidence. The main findings of this study were as follows: (a) women’s eating habits changed during pregnancy, with a tendency to a healthier diet that followed current recommendations; (b) 2% reported restricting their diet during pregnancy and; (c) most of the women did not take frequent measurements of their bodies and did not count calories consumed during pregnancy. In total, 10.71% of YG women and 13.33% of OG women took frequent body measurements and counted calorie intake. Most women changed their eating habits during pregnancy to healthier ones compared to before pregnancy. During pregnancy, the respondents (n= 74) mainly ate five meals a day. The amount of fruit and vegetables consumed in their diet was 80.20% (n = 81) and 84.16% (n = 85). Respondents reduced their coffee intake during pregnancy, while they increased their water intake to about 2 L a day. Most pregnant women used vitamin and mineral supplements. This indicates a high self-awareness among respondents on the topic of nutrition during pregnancy. In the survey, about 2% of respondents followed a weight-loss diet during pregnancy.

A study conducted by Gerontidis et al. on a group of 157 pregnant women showed that 61.1% of respondents (n = 96) improved their eating habits during pregnancy to more health-promoting ones. Egg consumption was declared by 93% (n = 146) of respondents, dairy consumption by 95.5% (n = 150) of pregnant women and fish consumption by 80.9% (n = 127) of respondents. Vitamin D supplementation was declared by 40.1% (n = 63) of subjects, while folic acid supplementation was declared by 87.9% (n = 138) [24].

A study conducted by Corrales-Gutierrez analyzed the effect of pregnant women’s education on their eating habits. Younger women and those with lower levels of education consume fewer fruits, vegetables and whole-grain breads, while on the other hand, they drink more caffeine-rich beverages. The level of physical activity in pregnant women was also studied. In this case as well, a low level of education was associated with a lower frequency of its practice [25].

Pregnancy can be the beginning of beneficial changes in eating habits in the long term. On the other hand, the desire to improve one’s lifestyle can also lead to various disorders such as over-controlling one’s weight, dissatisfaction with weight gain, the onset of depressive states and the onset of pregorexia.

Women’s management of weight gain during pregnancy often leads to dietary restriction. In our study, three (2.97%) pregnant women counted the number of calories they consumed per day, and two (1.98%) followed a weight-loss diet during pregnancy. It is well known that gaining weight during pregnancy contributes to postpartum women’s body weight retention. The 2.97% of women had a desire to return to their pre-pregnancy weights, similar to the findings of another study by Shiraishi et al. of pregnant Japanese women. A study found that 12.9% of Japanese women consumed insufficient food during pregnancy [26].

Pregnancy is associated with weight changes. In this study, 72,22% (n = 78) of respondents specified pregnancy as a positive period in their lives. The women’s self-esteem was good and very good for 45.54% (n = 46), with a high confidence in 50.49% (n = 51) and attractiveness being described as good and very good by 41.58% (n = 42). This is the same as in a study by Kostecka J. et al., where these changes were considered normal by 92% of the women surveyed, while 1/3 of pregnant women were uncomfortable with these changes. The study analyzed the mood, self-confidence and attractiveness of pregnant women. The largest percentage were women with their first pregnancy (n = 63; 62.38%), who rated their well-being to the greatest extent, describing it as “well” (n = 18; 28.57%) and “maybe” (n = 17; 26.98%). Analyses of the issues presented in terms of the number of pregnancies experienced showed no statistical significance [27].

Age can also be an important factor in how pregnant women feel. A study by Brunton [28] suggests that pregnancy at a more advanced age may be associated with more anxiety about the baby and thus negatively affect psychological aspects of the woman.

Harrison et al. reported that while the symptoms of eating disorders subside or diminish after pregnancy, their severity can be observed after childbirth [29]. Lee et al. indicated that changes in body image during pregnancy are more socially acceptable, while it is more difficult to accept a woman’s appearance after delivery [30].

A study in Lebanon showed that women’s dissatisfaction with their appearance during pregnancy was most often due to the influence of the media and pregnant celebrities. This promoted the development of various eating disorders during pregnancy. Lower socioeconomic status and polygamy influenced more favorable eating habits and a lower risk of developing eating disorders [31].

While the primary aim of this study was to analyze eating habits, body perception, attractiveness and self-confidence during pregnancy, the inclusion of parity as a variable in relation to psychological well-being emerged as an additional exploratory analysis. Although the results indicated no significant statistical relationship between the number of previous pregnancies and the sense of self-confidence (*p* = 0.98), this aspect was not initially defined as a core focus of the study. Future research could further explore this dimension to determine its potential impact on maternal psychological health during pregnancy.

The conducted study provides valuable information; however, the aspect of the impact of physical activity on the mental and physical health of pregnant women should be considered in the future. It seems interesting to expand the study to analyze in detail the effects of specific dietary components, such as vitamins, minerals, or nutrients on the mental health of pregnant women, and to examine the long-term effects of changes in pregnant women’s eating habits on their mental health, as well as their ability to return to a healthy lifestyle after giving birth. The study indicates that many women change their eating habits to healthier ones during pregnancy. Therefore, it is important to promote healthy eating habits among moms-to-be as early as the planning stage of pregnancy and to encourage the consumption of more vegetables, fruits and fluids.

Women should have knowledge about important supplements for them to take during pregnancy like folic acid, iron, calcium or omega-3 fatty acids. It is worth encouraging pregnant women to take vitamin and mineral supplements to ensure an adequate supply of essential nutrients for the development of the baby and the maintenance of the mother’s health.

Weight-loss diets during pregnancy should be warned against, as they can adversely affect the health of the mother and baby. Instead, a healthy and balanced diet should be promoted. It is important to point out to women the importance of regularly monitoring body circumference and energy intake during pregnancy, which can help maintain a healthy lifestyle and an adequate caloric supply for the baby’s development. The study found that pregnant women’s eating habits and well-being were not significantly associated with each other, regardless of age. Therefore, it is important to provide support for pregnant women in maintaining good mental health by promoting physical activity, ensuring an adequate diet and providing social support.

### Strengths and Weaknesses of the Study

The strengths of the study include the fact that the mental health aspect of pregnant women is a highly relevant topic due to the changing bodies of women, so the research presented here addresses an important topic and provides valuable results that can provide comparative value.

The study also has some limitations. The women who were surveyed mainly had a university education and very good/good material conditions. It would be worthwhile to expand the study group and diversify it more to assess causal mechanisms realistically. It is worth noting that while purposive sampling can provide insightful information about a specific group, the survey results may have some limitations in terms of generalizing to the entire population. The limited sample size of only 101 women, collected over a span of two years, represents a significant limitation of the study, which may affect the generalizability of the results. Additionally, the use of the CAWI method, although acceptable in psychological research, may introduce biases related to self-selection and the lack of control over the environment in which respondents complete the surveys.

It would be worth leaning harder into the study’s methodology and the group studied. It would be useful to expand the study in terms of pregnant women with different levels of education and to analyze whether education improves/changes eating habits. The age of the women and the correlation between pregnant women’s age and eating habits could also be an important element. Psychological aspects and the correlation of whether age and education affect the self-perception and well-being of pregnant women may also be of interest.

Purposive sampling has limitations; people who are members of Facebook groups had priority in finding out about the survey. This is an important limitation of the survey, as people who do not have this platform were restricted from accessing the survey and taking part in the study. However, the use of purposive sampling was important to achieve the goals of the survey and to ensure that the results were representative of the defined population.

In addition, the verification of participants’ email addresses was used to eliminate potential duplication of surveys. Each participant was required to provide a unique email address, which allowed us to identify and verify whether the individual had already completed the survey. To increase the effectiveness of eliminating duplicate responses, it would have been necessary to use IP address verification, but the Google Forms platform we chose does not have this capability.

It is worth extending the study in terms of direct contact with probands, such as in a birthing school, a gynecologist’s office or an obstetrician’s office. It is also necessary to increase the size of the study group. The simplicity of the statistical methods used is a result of the relatively small sample size of 101 participants, which limited the feasibility of applying more complex statistical analyses.

## 5. Conclusions

The present study showed that pregnant women try to improve their eating habits during pregnancy. This indicates a high self-awareness of respondents on the topic of nutrition during pregnancy. There is no significant effect of the number of pregnancies on mood, attractiveness and self-confidence in pregnant women. Also, there was no significant relationship between eating habits and mood in pregnant women, regardless of their age. Respondents who were more likely to take body circumference measurements equally often monitored their daily energy intake. The women’s eating habits and well-being were also not affected by their age.

Compiling information on women’s eating habits during pregnancy can help health care professionals to prioritize health care and support women in making informed choices. The pregnancy period is an opportunity to encourage a healthy lifestyle for women with regard to eating habits, as well as to promote proper eating habits and their maintenance after pregnancy. Unfortunately, many eating habits may revert to pre-pregnancy status due to lack of time, fatigue and support from partners and family. In addition to routine care and pregnant women, health care workers should set aside time to recommend healthy lifestyles.

## Figures and Tables

**Figure 1 healthcare-12-01932-f001:**
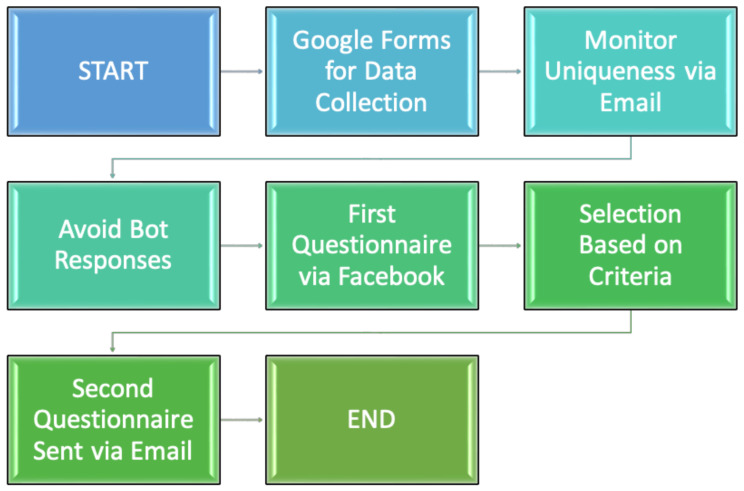
Flowchart of data collection process.

**Table 1 healthcare-12-01932-t001:** Characteristics of the study group by age groups (n = 101).

	Variables	Age [Years]	Height [cm]	Pre-Pregnancy Weight [kg]	Body Weight—1st Trimester [kg]	Body Weight—2nd Trimester [kg]	Body Weight—3rd Trimester [kg]
Total	X	30.01	165.52	63.4	65.12	70.35	77.4
	SD	4.75	5.29	13.23	12.66	12.29	13.93
YG (n = 56)	X	26.75	165.29	65.38	66.63	72.21	79.94
	SD	2.85	5.29	13.94	13.15	12.72	14.93
OG (n = 45)	X	34.1	165.73	60.93	63.24	68.03	74.24
	SD	3.29	5.2	11.84	11.76	11.31	11.85

YG—younger group (18–30 years), OG—older group (31–49 years), X—average, SD—standard deviation.

**Table 2 healthcare-12-01932-t002:** Comparison of women’s eating habits before pregnancy and during pregnancy.

	Before Pregnancy (n = 101)		During Pregnancy (n = 101)		*p*-Value
Eating Habits	Yes	No	Yes	No	
Eating 3 meals	50 (49.50%)	51 (50,50%)	37 (36.63%)	64 (63.37%)	*p* = 0.01 *
Eating 5 meals	46 (45.54%)	55 (54.45%)	74 (73.27%)	27 (26.73%)	*p* = 0.00 *
Eating breakfast regularly	53 (52.47%)	48 (47.52%)	89 (88.12%)	12 (11.88%)	*p* = 0.00 *
Daily consumption of vegetables	59 (58.42%)	42 (41.58%)	81 (80.20%)	20 (19.80%)	*p* = 0.00 *
Daily consumption of fruit	59 (58.42%)	42 (41.58%)	85 (84.16%)	16 (15.84%)	*p* = 0.00 *
Consuming 2 L of water daily	36 (35.64%)	65 (64.36%)	69 (68.32%)	32 (31.68%)	*p* = 0.00 *
Choosing quality foods	46 (45.54%)	55 (54.45%)	65 (64.36%)	36 (35.64%)	*p* = 0.00 *
Daily consumption of sweets	58 (57.43%)	43 (42.57%)	40 (39.60%)	61 (60.40%)	*p* = 0.00 *
Limiting the consumption of fast-food	49 (48.51%)	52 (51.48%)	65 (64.36%)	36 (35.64%)	*p* = 0.01 *
Food hygiene—eating meals in peace	42 (41.58%)	59 (58.42%)	88 (87.13%)	13 (12.87%)	*p* = 0.00 *
Food hygiene—eating meals on the run	66 (65.35%)	35 (34.65%)	15 (14.85%)	86 (85.15%)	*p* = 0.00 *

* *p* < 0.05.

**Table 3 healthcare-12-01932-t003:** Weight gain in women during pregnancy compared to baseline BMI by age group (n = 101).

BMI before Pregnancy						
			Women Underweight before Pregnancy (BMI ≤ 18.5)	Women with Normal BMI before Pregnancy (BMI: 18.4–24.99)	Overweight Women before Pregnancy (BMI: 25–29.99)	Women in Obesity before Pregnancy (BMI ≥ 30)
Weight gain during pregnancy	too low (n = 32)	Total (n = 32)	3 (9.7%)	25 (78.13%)	1 (3.13%)	3 (9.38%)
		YG (n = 19)	3 (15.79)	14 (73.68%)	1 (5.26%)	1 (5.26%)
		OG (n = 13)	0	11 (84.61%)	0	2 (15.38%)
		*p*-value	*p* = 0.12	*p* = 0.96	*p* = 0.37	*p* = 0.45
	correct (n = 37)	Total (n = 37)	4 (10.81%)	21 (56.76%)	9 (24.32%)	3 (8.11%)
		YG (n = 18)	2 (11.11%)	8 (44.44%)	6 (33.33%)	2 (11.11%)
		OG (n = 19)	2 (10.53%)	13 (68.42%)	3 (15.79%)	1 (5.26%)
		*p*	*p* = 0.83	*p* = 0.15	*p* = 0.52	*p* = 0.70
	too high (n = 32)	Total (n = 32)	6 (18.75%)	10 (31.25%)	13 (40.63%)	3 (9.38%)
		YG (n = 19)	3 (15.79%)	3 (15.79%)	10 (52.63%)	3 (15.79%)
		OG (n = 13)	3 (23.08%)	7 (53.85%)	3 (23.08%)	0
		*p*-value	*p* = 0.79	*p* = 0.12	*p* = 0.14	*p* = 0.12
	Total (n = 101)		13 (12.87%)	56 (55.45%)	23 (22.77%)	9 (8.91%)

YG—younger group (18–30 years), OG—older group (31–49 years).

**Table 4 healthcare-12-01932-t004:** Self-esteem, attractiveness and self-confidence of pregnant women according to the number of past pregnancies.

Variable	Pregnancy	Very Bad Scale-1	Bad Scale-2	Could Be Scale-3	Good Scale-4	Very Good Scale-5	*p*-Value
Self-esteem of pregnant women	First	4 (6.35%)	13 (20.63%)	17 (26.98%)	18 (28.57%)	11 (17.46%)	*p* = 0.99
	Second	6 (20.0%)	4 (13.33%)	7 (23.33%)	8 (26.67%)	5 (16.67%)	
	Third	1 (25.0%)	2 (50.0%)	0 (0.0%)	0 (0.0%)	1 (25.0%)	
	Fourth and subsequent	1 (25.0%)	0 (0.0%)	0 (0.0%)	2 (50.0%)	1 (25.0%)	
Attractiveness of pregnant women	First	7 (11.11%)	12 (19.05%)	18 (28.57%)	16 (25.40%)	10 (18.57%)	*p* = 0.99
	Second	6 (20.0%)	8 (26.67%)	6 (20.0%)	6 (20.0%)	4 (12.12%)	
	Third	0 (0.0%)	1 (25.0%)	1 (25.0%)	1 (25.0%)	1 (25.0%)	
	Fourth and subsequent	0 (0.0%)	0 (0.0%)	0 (0.0%)	3 (75.0%)	1 (25.0%)	
Self-confidence of pregnant women	First	5 (7.94%)	13 (20.63%)	13 (20.63%)	18 (28.57%)	14 (22.22%)	*p* = 0.98
	Second	7 (23.33%)	3 (10.0%)	7 (23.33%)	8 (26.67%)	5 (16.67%)	
	Third	0 (0.0%)	1 (25.0%)	0 (0.0%)	2 (50.0%)	1 (25.0%)	
	Fourth and subsequent	0 (0.0%)	0 (0.0%)	1 (25.0%)	0 (0.0%)	3 (75.0%)	

## Data Availability

The datasets used and/or analyzed during the current study are available from the corresponding author on reasonable request.

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
