# Peer review of "Eating Habits Related to Pregnancy, Body Perception, Attractiveness and Self-Confidence"

_healthcare, 2024, doi:10.3390/healthcare12191932_

Round 1

Reviewer 1 Report

Comments and Suggestions for Authors

Dear authors/Editor, thank you for inviting me to review the manuscript: The manuscript, titled “Effects of Pregnancy on Eating Habits, Body Perception, Attractiveness, and Self-Confidence,” provides valuable insight into dietary and cognitive changes during pregnancy. The study design, which includes pre-and post-pregnancy surveys, effectively captures changes in eating habits and body image. The use of a questionnaire to collect data is appropriate, and the longitudinal aspect provides a comprehensive overview of dietary changes and their effects.

The results suggest a shift toward healthier dietary practices, including increased vegetable and fruit intake and reduced coffee consumption, consistent with existing literature on dietary changes during pregnancy. However, the finding that only 2% of respondents followed a weight loss diet and the lack of significant associations between eating habits and well-being suggest that the study could benefit from a broader analysis of psychological factors or larger sample size to improve generalizability.

Overall, the manuscript provides useful information regarding dietary changes during pregnancy but would benefit from a deeper exploration of psychological dimensions and a more detailed discussion of limitations.

With regards,

Reviewer

Author Response

Thank you so much for taking the time to evaluate our work. We have tried to incorporate all your valuable suggestions. If we could improve our work in any way, please let us know.

Comment 1

The results suggest a shift toward healthier dietary practices, including increased vegetable and fruit intake and reduced coffee consumption, consistent with existing literature on dietary changes during pregnancy. However, the finding that only 2% of respondents followed a weight loss diet and the lack of significant associations between eating habits and well-being suggest that the study could benefit from a broader analysis of psychological factors or larger sample size to improve generalizability.

Overall, the manuscript provides useful information regarding dietary changes during pregnancy but would benefit from a deeper exploration of psychological dimensions and a more detailed discussion of limitations.

We sincerely thank you for your guidance. Changes have been made in the manuscript.

Thank you for your help. Your guidance is invaluable.

Kind regards,

Authors.

Reviewer 2 Report

Comments and Suggestions for Authors

Dear Author,

The content of your manuscript was intriguing. However, since you have categorized pregnant women in your study, I believe you should discuss more about comparing data across different ages, even if there were no significant differences. Please include this aspect in your discussion. As mentioned in the paper, your subjects are biased towards those with higher education. It is likely important to explore how pregnant women with lower education, lower income, and limited access to information experience changes due to pregnancy. I hope you will aim to gather data from a more varied group of subjects in the future.

 Sincerely,

Author Response

Thank you so much for taking the time to evaluate our work. We have tried to incorporate all your valuable suggestions. If we could improve our work in any way, please let us know.

Comment 1

The content of your manuscript was intriguing. However, since you have categorized pregnant women in your study, I believe you should discuss more about comparing data across different ages, even if there were no significant differences. Please include this aspect in your discussion. As mentioned in the paper, your subjects are biased towards those with higher education. It is likely important to explore how pregnant women with lower education, lower income, and limited access to information experience changes due to pregnancy. I hope you will aim to gather data from a more varied group of subjects in the future.

We plan to expand this study to include various aspects of education and the age of pregnant women. We thank you very much for your guidance. Your guidance is invaluable.

Kind regards,

Authors.

Reviewer 3 Report

Comments and Suggestions for Authors

Review: healthcare-3131873

Title: Effects of pregnancy on eating habits, body perception, attractiveness and self-confidence

Dear authors,

Thank you for the time and effort in conducting the study “Effects of pregnancy on eating habits, body perception, attractiveness and self-confidence”.

The manuscript has a cross-sectional design and addresses dietary habits among pregnant women between January 2021 and December 2022 (N =101). The authors aimed to evaluate changes in women's eating habits before and during pregnancy and to analyze the impact of pregnancy on women's body perception and attractiveness, and self-confidence.  The authors also aimed to evaluate whether women's age served as a determinant for the above-mentioned aim.

Please find my comments below, outlined per section in the manuscript.

Title

The title imply that this is an association study, but it is not. Perhaps pregnancy related habits or alike would be more appropriate.

Abstract

The abstract is structured and has the necessary parts, however – of importance. In the method part: the inclusion of the aim would be preferable as would briefly mention the statistical method of choice for the main analyses presented in the result section of the abstract. For the conclusion, perhaps add some clinical implications. What do the women benefit from this study?

Keywords

Perhaps shift them into other keywords that are not in the title or abstract to increase searchability

Introduction

The introduction lacks clarity and a natural link to the aim of the manuscript. It comes about quite unstructured and perhaps that is a reflection of the pathways between aim -> hypothesis -> methos -> analyses -> results -> discussion -> conclusion which is a bit unpolished.  What is missing is a narrowed review of the field, what is known, what has been done on the topic, stemming from the purpose of the study.

Page 1 line 39. The use of reference 6 is slightly unclear, why were that recommendation used? The reference is not in English making it impossible for me and other international readers to understand it. Are those standars perhaps in relation to WHO?

Page 2 start line 56. Remove the table from the introduction and have the information intext only

Page 2 line 57-58. The say that womens attractiveness is influenced by her weight. Perhaps a more appropriate approach would be to introduce something about weight gain and pregnancy related risks? In the reference [12] the authors talks about that.

Page 2 line 67-75. This section talks about eating disorders which is not part of the study aim. Remove this and if it is important for your findings, move to the discussion.

Page 2 line 76-79. The aim of this study is to a) evaluate changes in eating habits prior to or during pregnancy. b) how the pregnancy per se impacts the women’s self-view and perceptive attractiveness, and confidence. c) if the eating habits, self-view, attractiveness and confidence is driven by age. There are a few concerns here. A change in eating habits could be driven by so many other things that just age, and that could be a first line of adjustment such as cravings, recommendation and restrictions from the midwife and how the women feel during pregnancy such as morning sickness. Come to think of it, is this accounted for in the introduction?

Page 2 line 70-84. The hypothesis might be a little unnecessary.

Methods

Moreover, the method is quite unstructured perhaps divide the sections and add valuable titles. Such as Subjects, data collection and measurement.

Page 2 line 87. The sample size is very limited with a total of 101 women. One concern is that the data collection forth went from January 2021 to December 2022 yet still only rendered 101 women. This needs to be highlighted in the discussion part.

 Page 2 line 87-90. From A CAWI to psychological research. This could be moved to the discussion where a more thorough discussion about the method is needed.

Page 2 line 90. The authors refer to a questionnaire. Name the questionnaire and provide a ref to the author of the questionnaire also provide reliability scores if available. Page 3 line 99. Make sure to not use a language such as that google platform was used due to availability.

Page 3 line 100- onwards. This is more of a discussion. Try to make it more clear how and where you collected the data as for now it is to hard for the reader to follow the process. I highly recommend the authors to use a Flow chart of the process.

Page 3 line 114. Is the survey anonymous or did the authors work under confidentiality? As I understood information was gathered.

Page 3 line 113. Perhaps ad a title as Ethical consideration and describe your ethical concerns there such as informed consent etc.

Page 3 starting line 123. This is the measurements. Please provide titles of the different measurements used, provide information on the questionnaires but also reliability estimates and such.

Just to add: Please use “corresponded to” instead of “meant”

Make sure to have a more describing approach in the methods, like a receipt so that other can replicate. All other information is to be moved to the discussion.

Page 4 line 147. The authors have an indication of malnutrition, is that something that WHO actually uses for normal BMI?

Page 4 line 155-160. Regarding the inconclusion and exclusions. Inclusion 6 and 7 can be moved, that is not study inclusions. Participants will be in the study and accounted for even if the questionnaire is not fully answered, it is unethical to force a subject to answer all the study questions. You can exclude them within the analysis due to missing data but that is something else. The exclusions are the same as inclusion… Doesn’t make sense.

Page 4 line 162. The formulation about the chosen methods should be more straight forward. There is no need to explain the use of any basic method here wrt normality for example.

Is the statistical method perhaps a little simple? Is that due to the sample size? A discussion of this in the discussion would be preferable.

Results

The results are a little bit messy. Please use subtitles and start with “Sample characteristics” and perhaps make it as Table 1, meaning expand table 1 which is now only a description of BMI and not to much of a sample characteristics - text with a lot of numbers is hard to read.

Page 5 Table 1. There is a note of Me, Min and Max, where is it in the table?

The first and foremost aim is to evaluate eating habits before and during pregnancy. The results section should start with the main findings (after sample characteristics).

Page 5 line 188. The numbers doesn’t add up you have 101.2% and 102 participants.

Page 5 line 188 onwards. There is a lot of results on weight gain, which is not part of the aim?

Page 5 line 195. Eating habits, if the aim is to evaluate a change in eating habits the results should be according to that. Please state if or if not, there was a significant change in eating habits over the two time points. Characteristics of before and during doesn’t say much if the change is real.

Page 6 line 211. These numbers are duplicated with table 4. It is not kutym to duplicate in text with table, it should be in either.

Page 6 line 211. “Breakfast was eaten with regularity between meals” what does that mean?

Page 6 line 219. Please present the accurate p-values in the table if <0.05.

Page 7 line 229. Table 5. Perhaps the table makes the findings hard to access instead of making it easier. Provide the findings in text only.

Page 7 line 231. There is some text here that is for the method section. Keep results section clean and only present findings.

Page 7 line 246 Table 6. The talk about parity surprised me as a reader. Was that in your aim?

Discussion

Now as I read the first line of the main findings of the discussion, I feel that it would be good if eating habits are defined in the method. What is it here in the study?

I will provide general comments of the discussion instead of line wise.

-          Make sure to use another language than in the results section, otherwise it will be only a repetition of the findings.

-          Try to emphasize what you found and how that can be related to previous studies. There are several lines in the discussion that actually are really good but as an introduction to the study.

-          Try to present what your findings mean – if you found that 2.97% of the study women count their calorie intake, what does it mean? What could be a possible reason for that behavior?

-          There is a discussion of vitamin intake and other supplements. This must be clearer from the beginning why it is talked so much about. Couldn’t that just be a recommendation for all pregnant women?

-          Strength and limitation. The study population is not a strength. The measure of two time points is not a strength, its your study design?

-          There needs to be a discussion about the methodology, about the study population, about generalizability, about the use of the findings, about the clinical implications, where does this study fit within the field? etc.

-          The conclusion should be written in a way that you conclude the whole study and link it to a larger perspective and end with to whom it would benefit.

Author Response

Thank you so much for taking the time to evaluate our work. We have tried to incorporate all your valuable suggestions. If we could improve our work in any way, please let us know.

Comment 1

The title imply that this is an association study, but it is not. Perhaps pregnancy related habits or alike would be more appropriate.

Thank you for your guidance. The title has been changed.

Comment 2
The abstract is structured and has the necessary parts, however – of importance. In the method part: the inclusion of the aim would be preferable as would briefly mention the statistical method of choice for the main analyses presented in the result section of the abstract. For the conclusion, perhaps add some clinical implications. What do the women benefit from this study?

Thank for your guidance. Changes have been made to the text.

Comment 3

Keywords

Perhaps shift them into other keywords that are not in the title or abstract to increase searchability.

Thank for your guidance. Changes have been made to the text.

Comment 4

Page 1 line 39. The use of reference 6 is slightly unclear, why were that recommendation used? The reference is not in English making it impossible for me and other international readers to understand it. Are those standars perhaps in relation to WHO?

The article is by Professor Dorota Szostak-Węgierek and deals with the principles of nutrition for pregnant women. We have also added recommendations from the WHO, which are analogous and coincide with the Polish recommendations.

Comment 5

Page 2 start line 56. Remove the table from the introduction and have the information intext only.

Thank for your guidance. Changes have been made to the text.

Comment 6

Page 2 line 57-58. The say that womens attractiveness is influenced by her weight. Perhaps a more appropriate approach would be to introduce something about weight gain and pregnancy related risks? In the reference [12] the authors talks about that.

Thank for your guidance. Changes have been made to the text.

Comment 7

Page 2 line 67-75. This section talks about eating disorders which is not part of the study aim. Remove this and if it is important for your findings, move to the discussion.

Thank for your guidance. Changes have been made to the text.

Comment 8

Page 2 line 76-79. The aim of this study is to a) evaluate changes in eating habits prior to or during pregnancy. b) how the pregnancy per se impacts the women’s self-view and perceptive attractiveness, and confidence. c) if the eating habits, self-view, attractiveness and confidence is driven by age. There are a few concerns here. A change in eating habits could be driven by so many other things that just age, and that could be a first line of adjustment such as cravings, recommendation and restrictions from the midwife and how the women feel during pregnancy such as morning sickness. Come to think of it, is this accounted for in the introduction?

Thank for your guidance. Changes have been made to the text.

Comment 9

Page 2 line 70-84. The hypothesis might be a little unnecessary.

Thank for your guidance. Changes have been made to the text.

Comment 10

Moreover, the method is quite unstructured perhaps divide the sections and add valuable titles. Such as Subjects, data collection and measurement.

Thank for your guidance. Changes have been made to the text.

Comment 11

Page 2 line 87. The sample size is very limited with a total of 101 women. One concern is that the data collection forth went from January 2021 to December 2022 yet still only rendered 101 women. This needs to be highlighted in the discussion part.

Thank for your guidance. Changes have been made to the text.

Comment 12

 Page 2 line 87-90. From A CAWI to psychological research. This could be moved to the discussion where a more thorough discussion about the method is needed.

Thank for your guidance. Changes have been made to the text.

Comment 13

Page 2 line 90. The authors refer to a questionnaire. Name the questionnaire and provide a ref to the author of the questionnaire also provide reliability scores if available. Page 3 line 99. Make sure to not use a language such as that google platform was used due to availability.

Thank for your guidance. Changes have been made to the text. The questionnaire is described in section 2.2 Research tools.

Comment 14

Page 3 line 100- onwards. This is more of a discussion. Try to make it more clear how and where you collected the data as for now it is to hard for the reader to follow the process. I highly recommend the authors to use a Flow chart of the process.

Added figure 1 as requested by the reviewer.

Comment 15

Page 3 line 114. Is the survey anonymous or did the authors work under confidentiality? As I understood information was gathered.

The information was collected under conditions of confidentiality.

Comment 16

Page 3 line 113. Perhaps ad a title as Ethical consideration and describe your ethical concerns there such as informed consent etc.

Thank for your guidance. Changes have been made to the text.

Comment 17

Page 3 starting line 123. This is the measurements. Please provide titles of the different measurements used, provide information on the questionnaires but also reliability estimates and such.

Thank for your guidance. Changes have been made to the text.

Comment 18

Just to add: Please use “corresponded to” instead of “meant”

Thank for your guidance. Changes have been made to the text.

Comment 19

Make sure to have a more describing approach in the methods, like a receipt so that other can replicate. All other information is to be moved to the discussion.

Thank you for the valuable tip, we have revised the text for it.

Comment 20

Page 4 line 147. The authors have an indication of malnutrition, is that something that WHO actually uses for normal BMI?

Thank you for your valuable comment, in accordance with WHO's recommendations we have updated the description.

Comment 21

Page 4 line 155-160. Regarding the inconclusion and exclusions. Inclusion 6 and 7 can be moved, that is not study inclusions. Participants will be in the study and accounted for even if the questionnaire is not fully answered, it is unethical to force a subject to answer all the study questions. You can exclude them within the analysis due to missing data but that is something else. The exclusions are the same as inclusion… Doesn’t make sense.

Thank you very much for your attention, of course, changed in the text.

Comment 22

Page 4 line 162. The formulation about the chosen methods should be more straight forward. There is no need to explain the use of any basic method here wrt normality for example.

Thank for your guidance. Changes have been made to the text.

Comment 23

Is the statistical method perhaps a little simple? Is that due to the sample size? A discussion of this in the discussion would be preferable.

An appropriate description has been added.

Comment 24

The results are a little bit messy. Please use subtitles and start with “Sample characteristics” and perhaps make it as Table 1, meaning expand table 1 which is now only a description of BMI and not to much of a sample characteristics - text with a lot of numbers is hard to read.

Thank for your guidance. Changes have been made to the text.

Comment 25

Page 5 Table 1. There is a note of Me, Min and Max, where is it in the table?

The table does not contain the indicated parameters, changes have been made to the table description.

Comment 26

The first and foremost aim is to evaluate eating habits before and during pregnancy. The results section should start with the main findings (after sample characteristics).

Changed as suggested..

Comment 27

Page 5 line 188. The numbers doesn’t add up you have 101.2% and 102 participants.

Thank for. Changes have been made to the text.

Comment 28

Page 5 line 188 onwards. There is a lot of results on weight gain, which is not part of the aim?

We think this is an important topic that we wanted to address as well a this study. We have added an appropriate objective.

Comment 29

Page 5 line 195. Eating habits, if the aim is to evaluate a change in eating habits the results should be according to that. Please state if or if not, there was a significant change in eating habits over the two time points. Characteristics of before and during doesn’t say much if the change is real.

The results have been corrected as suggested.

Comment 30

Page 6 line 211. These numbers are duplicated with table 4. It is not kutym to duplicate in text with table, it should be in either.

The results have been corrected as suggested.

Comment 31

Page 6 line 211. “Breakfast was eaten with regularity between meals” what does that mean?

The results have been corrected.

Comment 32

Page 6 line 219. Please present the accurate p-values in the table if <0.05.

The values were presented to 2 decimal places.

Comment 33

Page 7 line 229. Table 5. Perhaps the table makes the findings hard to access instead of making it easier. Provide the findings in text only.

Thank for your guidance. Changes have been made to the text.

Comment 34

Page 7 line 231. There is some text here that is for the method section. Keep results section clean and only present findings.

Thank for your guidance. Changes have been made to the text.

Comment 35

Page 7 line 246 Table 6. The talk about parity surprised me as a reader. Was that in your aim?

Mention of parity (number of previous pregnancies) and its impact on self-esteem, attractiveness and self-confidence in pregnant women was further analyzed. We have added a relevant description to the study's discussion.

“While the primary aim of this study was to analyze eating habits, body perception, attractiveness, and self-confidence during pregnancy, the inclusion of parity as a variable in relation to psychological well-being emerged as an additional exploratory analysis. Although the results indicated no significant statistical relationship between the number of previous pregnancies and the sense of self-confidence (p=0.98), this aspect was not initially defined as a core focus of the study. Future research could further explore this dimension to determine its potential impact on maternal psychological health during pregnancy.”

Comment 36

Now as I read the first line of the main findings of the discussion, I feel that it would be good if eating habits are defined in the method. What is it here in the study?

Thank for your guidance. Changes have been made to the text.

Comment 37

Make sure to use another language than in the results section, otherwise it will be only a repetition of the findings.

Thank for your guidance.

Comment 38

 Try to emphasize what you found and how that can be related to previous studies. There are several lines in the discussion that actually are really good but as an introduction to the study.

Thank for your guidance.

Comment 39

Try to present what your findings mean – if you found that 2.97% of the study women count their calorie intake, what does it mean? What could be a possible reason for that behavior?

Thank for your guidance. Changes have been made to the text.

Comment 40

 There is a discussion of vitamin intake and other supplements. This must be clearer from the beginning why it is talked so much about. Couldn’t that just be a recommendation for all pregnant women?

Thank for your guidance. Changes have been made to the text.

Comment 41

Strength and limitation. The study population is not a strength. The measure of two time points is not a strength, its your study design?

Thank for your guidance. Changes have been made to the text.

Comment 42

There needs to be a discussion about the methodology, about the study population, about generalizability, about the use of the findings, about the clinical implications, where does this study fit within the field? etc.

Thank for your guidance. Changes have been made to the text.

Comment 43

The conclusion should be written in a way that you conclude the whole study and link it to a larger perspective and end with to whom it would benefit.

Thank for your guidance. Changes have been made to the text.

Thank you for your help. Your guidance is invaluable.

Kind regards,

Authors.